# Lamb Wave Based Structural Damage Detection Using Stationarity Tests

**DOI:** 10.3390/ma14226823

**Published:** 2021-11-12

**Authors:** Phong B. Dao, Wieslaw J. Staszewski

**Affiliations:** 1Department of Robotics and Mechatronics, AGH University of Science and Technology, Al. Mickiewicza 30, 30-059 Krakow, Poland; w.j.staszewski@agh.edu.pl; 2School of Mechanical Engineering, Hanoi University of Science and Technology, 1 Dai Co Viet, Hai Ba Trung, Hanoi, Vietnam

**Keywords:** structural health monitoring, integrated piezoceramic transducers, Lamb waves, temperature variations, stationarity and nonstationarity, stationarity test, unit root test

## Abstract

Lamb waves have been widely used for structural damage detection. However, practical applications of this technique are still limited. One of the main reasons is due to the complexity of Lamb wave propagation modes. Therefore, instead of directly analysing and interpreting Lamb wave propagation modes for information about health conditions of the structure, this study has proposed another approach that is based on statistical analyses of the stationarity of Lamb waves. The method is validated by using Lamb wave data from intact and damaged aluminium plates exposed to temperature variations. Four popular unit root testing methods, including Augmented Dickey–Fuller (ADF) test, Kwiatkowski–Phillips–Schmidt–Shin (KPSS) test, Phillips–Perron (PP) test, and Leybourne–McCabe (LM) test, have been investigated and compared in order to understand and make statistical inference about the stationarity of Lamb wave data before and after hole damages are introduced to the aluminium plate. The separation between t-statistic features, obtained from the unit root tests on Lamb wave data, is used for damage detection. The results show that both ADF test and KPSS test can detect damage, while both PP and LM tests were not significant for identifying damage. Moreover, the ADF test was more stable with respect to temperature changes than the KPSS test. However, the KPSS test can detect damage better than the ADF test. Moreover, both KPSS and ADF tests can consistently detect damages in conditions where temperatures vary below 60 °C. However, their t-statistics fluctuate more (or less homogeneous) for temperatures higher than 65 °C. This suggests that both ADF and KPSS tests should be used together for Lamb wave based structural damage detection. The proposed stationarity-based approach is motivated by its simplicity and efficiency. Since the method is based on the concept of stationarity of a time series, it can find applications not only in Lamb wave based SHM but also in condition monitoring and fault diagnosis of industrial systems.

## 1. Introduction

It is well known that a damage detection process in the context of structural health monitoring (SHM) involves three stages [1,2]: (1) the acquisition of data from the structure of interest using periodically sampled dynamic response measurements; (2) the extraction of damage-sensitive features from the acquired data; and (3) the decision on the current state of the structure’s health based on the statistical analysis of damage-sensitive features. Lamb wave based SHM systems have been developed for structural damage detection for many years with the use of integrated, low-profile piezoceramic transducers, as discussed in [3,4,5,6,7,8,9,10,11,12]. One of the main problems with the method is that it is prone to contamination by changing environmental and operational conditions. Another major problem is the complexity of Lamb wave propagation, which is caused by two main reasons, as explained in [13,14,15]. First, an infinite number of different modes can simultaneously appear and propagate in a structure. Second, these modes are dispersive and overlap in both frequency and time domains. Many conventional signal processing methods has been applied to separate overlapping Lamb wave signals, as reported in [13], but the achieved results are not satisfactory. In order to avoid this problem, the current study has proposed another approach; that is, instead of directly using Lamb wave signals for damage detection, we should analyse and interpret the damage-sensitive features obtained by using an appropriate signal processing technique on the signals/responses. A potential method is the statistical analysis of stationarity (or nonstationarity) of Lamb waves.

In the recent years, the ideas of statistical (non)stationarity of data and its relations with damage or fault detection have become essential to SHM, and these are the main research topics of this study. It is well known that stationarity and nonstationarity are explained in the fields of econometrics and statistics by using models. However, in engineering fields, statistical moments are often used [8,16]. Basically, statistical properties (such as mean, variance, and autocorrelation) of a stationary time series are constant over time [16,17]. Thus, the initial assumption here is that it is possible to reduce the problem of determining whether a system or structure varies with time to the problem of checking if given signals (or measured data) are changing or varying with time. Based on this assumption, the key idea of data-driven SHM approaches is to decide if the measured data changes its nature when a damage state appears in the structure. It is, thus, assumed in this study that such a change can potentially alter the (non)stationarity of data. In a broader view, this is well known as the change-point detection problem.

The above discussion leads to the research question of this study: How can we measure (or calculate) the stationarity or nonstationarity of SHM data in general and Lamb wave signals in particular? The solution for this question relies on the fact that, in essence, the existence of unit roots can be the cause for nonstationarity in time series data [18]. Basically, a unit root process (also known as a difference-stationary process) is a stochastic trend in a time series [19,20]. A unit root time series exhibits a systematic pattern that is unpredictable. Then, it is assumed that the degree of stationarity (or nonstationarity) of a Lamb wave signal can be quantified by using a unit root test or a stationarity test. If this is the case then a structural damage can be detected by using stationary statistical characteristics of Lamb wave data. The issue is investigated in this paper with the focus on developing a new stationarity-based approach for Lamb wave based structural damage detection.

Our literature search reveals that there are four unit root tests commonly used in the field of econometrics. These include the following: (1) Augmented Dickey–Fuller (ADF) test [21,22]; (2) Phillips–Perron (PP) test [23]; (3) Kwiatkowski–Phillips–Schmidt–Shin (KPSS) test [24]; and (4) Leybourne–McCabe (LM) stationarity test [25,26]. However, it should be noted that only the ADF test has been investigated for SHM problems, as reported in the literature [8,16,17,27,28,29,30,31]. In particular, the previous studies in [8,27,28] employed a combination of the ADF test and cointegration approach [32,33] for damage detection using Lamb waves, in which cointegration was used for temperature effect removal. This fact might result in an assumption in the field of SHM that ADF is the best unit root test or even the only tool that can be applied for testing stationarity of data. Therefore, this study has investigated and compared these four popular unit root tests for SHM with specific applications towards structural damage detection based on Lamb waves. The proposed methodology is based on the analysis of stationary statistical characteristics of Lamb wave data. The method is experimentally validated with a practical case study, i.e., structural damage detection in aluminium plates using lamb wave data contaminated by temperature variations. To the best of authors’ knowledge, research on different unit root testing methods for potential applications in Lamb wave based SHM has not been previously investigated in the literature.

The remaining parts of the paper are organised as follows. Section 2 introduces the theoretical background including basic concepts, time series and stationarity, and unit root tests. Then, four popular unit root testing methods (i.e., ADF, PP, KPSS, and LM test) are shortly described. Section 3 presents a new Lamb wave based structural damage detection method, which is based on the concept of stationarity of time series. Lamb wave experimental data contaminated by varying temperatures are described in Section 4. The results of damage detection using the four unit root testing methods are then presented and discussed in Section 5. Finally, the paper is concluded in Section 6.

## 2. Background

### 2.1. Basic Concepts

This subsection briefly describes some basic concepts related to time series analysis used in this study.

Let $\left\{z_{t}\}=\{\ldots ,z_{t-1},z_{t},z_{t+1},\ldots \right\}$ denote a time series that is represented by a sequence of random variables indexed by time t.

A time series in general can be decomposed into three elements [19,20]:
Trends ($T_{t}$) representing long term movements in the mean;Seasonal effects ($i_{t}$) and cycles ($c_{t}$) representing cyclical fluctuations in the series;Residuals ($u_{t}$) representing random or systematic fluctuations.

A time series is called stationary if the fundamental form of the data generating process remains the same over time [19,20]. This is manifested in the moments of the process. For example, mean stationarity implies that the expected value of the process is constant over time; in other words, we have the following.
(1)$$E\left[z_{t}\right]=\mu \;\forall t\;\mathrm{where}\mu \mathrm{is}\mathrm{constant}$$

Similarly, variance stationarity means that the variance is stable.
(2)$$\mathrm{var}\left[z_{t}\right]=E\left[{\left(z_{t}-\mu \right)}^{2}\right]\equiv \sigma_{z}^{2}\;\forall t$$

Moreover, it is similar with respect to covariance stationarity:(3)$$\mathrm{cov}\left[z_{t},z_{t-s}\right]=E\left[\left(z_{t}-\mu )(z_{t-s}-\mu \right)\right]=\lambda_{s}\;\forall s$$
where $\lambda_{s}$ is independent of t and is called the autocovariance function.

The above discussion refers to a “weak” form of stationarity (that is, stationarity in the first and second moments). More specifically, a weakly stationary time series has constant mean and variance, and covariance is independent of time. A stricter form of stationarity requires that the joint probability distribution of time series $\left\{z_{1},z_{2},\ldots ,z_{t}\right\}$ is the same as that for $\left\{z_{1+s},z_{2+s},\ldots ,z_{t+s}\right\}$ for all t and s. It should be noted that, in general, people are interested in the weak form of stationarity, and this concept is also used in this paper.

### 2.2. Time Series and Stationarity

Time trend models are often used in economics to analyse and explain stationarity (or nonstationarity) of time series. In general, most time series data have the structure of an autoregressive (*AR*) process, which in its first-order form AR(1) is defined as follows [19,20]:(4)$$z_{t}=\phi z_{t-1}+\varepsilon_{t}$$
where $\varepsilon_{t}$ is an independent Gaussian white noise process with zero mean. Based on different values of coefficient ϕ, three types of time series can be distinguished [20]:
For $\left|\phi \right|<1$, we have a stationary time series, which appears jagged but always oscillates around the mean, as illustrated in Figure 1;For $\left|\phi \right|>1$, we have a nonstationary time series, which is smoother and finally explodes, as shown in Figure 2;For ϕ=1, we have a random walk time series, which moves up and down; it behaves as a nonstationary time series but slowly, as plotted in Figure 3.

Stationarity and nonstationarity of engineering data can be designated through statistical moments of the process [8]. A stationary process has time-invariant moments, whereas statistical moments of a nonstationary process exhibit time dependence. In practice, nonstationarity of a time series usually manifests itself in trends, cycles, random walks, or combinations of the three.

Most of (economic) time series as they occur in practice often show *trends* through their history. Here, a trend may be observed as a long-term increase or decrease in the level of the time series, which can exist in either the mean, the variance, or both. Time series analysis and forecasting methods are concerned with making nonstationary series stationary by identifying and removing trends and seasonal effects. In other words, it is possible to stationarise a nonstationary time series by detrending. This can be performed by fitting linear trend lines to data and then subtracting them out prior to fitting a model. In this case, the time series is said to be *trend stationary* [19]. However, in most of the cases detrending is not often sufficient to make a nonstationary series turn stationary. In this situation, a common solution is to transform a nonstationary time series into a series of period-to-period or season-to-season differences. If this transformation produces a stationary series, the original series is said to be *difference stationary* [19].

Then, a key question here is how to test for (non)stationarity of a time series. Typically, data acquired from engineering systems or structures involve mostly two types of signals: (1) stationary time series (if $\left|\phi \right|<1$) and (2) nonstationary random walk series (if ϕ=1). Then, the concern would be if a unit root exists (ϕ=1) or does not exist ($\left|\phi \right|<1$). If a unit root exists, then one has a nonstationary time series; in the case where there is no unit root, then one has a stationary time series [18]. This can be performed by means of a unit root test, which makes statistical inference about the (non)stationarity of time series data.

### 2.3. Unit Root Tests

A unit root process (or a difference-stationary process) is a stochastic trend in a time series [19,20]. A unit root problem involves the existence of characteristic roots of a time series model on the unit circle. The existence of unit roots is the cause for nonstationarity in time series data [18]. Under the presence of unit roots, the standard distribution theory is not valid; for example, *t*-ratios will not follow a *t*-distribution. The presence of unit roots can result in serious issues for time series analysis, such as spurious regressions where one could obtain high *r*-squared values even if the data are uncorrelated. Thus, it is important to test for stationarity of the time series prior to any estimation in order to use the appropriate procedures for detrending.

Consider a time series $y_{t}$ in the form of trend-cycle decomposition:(5)$$y_{t}=TD_{t}+z_{t}$$
(6)$$TD_{t}=c+\delta t$$
where $z_{t}$ is an AR(1) process provided by Equation (4), $TD_{t}$ is a deterministic linear trend, and c and δ are constants and known as the drift and deterministic trend coefficients, respectively. Then, there are two possible cases to be considered:
If ϕ=1, then $z_{t}=z_{t-1}+\varepsilon_{t}=z_{0}+\sum_{i=1}^{t}\varepsilon_{i}$, where $\varepsilon_{t}$ is a white noise process, and so $y_{t}$ is a nonstationary random walk time series with a drift that contains *a stochastic trend or a unit root* (which is represented by the integrated sum $\sum_{i=1}^{t}\varepsilon_{i}$). In this case, shocks have permanent effects, and the time series shows an unpredictable pattern (see Figure 4). It is noted that process $y_{t}$ in this case is referred to as *difference stationary*, which is discussed in Section 2.2.If $\left|\phi \right|<1$, then $y_{t}$ is a stationary time series around the deterministic linear trend $TD_{t}$. In this case, shocks have transitory effects, as illustrated in Figure 5. The process $y_{t}$ in this case is referred to as *trend stationary*, which is mentioned in Section 2.2.

In both examples presented in Figure 4 and Figure 5, two identical time series $x_{t}$ and $y_{t}$ were used; however, the shock was only inserted into the time series $y_{t}$ at data sample 50. In addition, Figure 5 shows that both time series $x_{t}$ and $y_{t}$ followed a deterministic linear trend $TD_{t}=0.1t$, where t is discrete points of time. In this case, $y_{t}$ is a trend-stationary process, and it can be observed in Figure 5 that the shock has transitory effects on $y_{t}$, and the series could return to its original path and then continue by varying around the trend as the same series $x_{t}$. However, if $y_{t}$ is a unit root process, one can observe in Figure 4 that the shock has permanent effects on $y_{t}$; therefore, the series could not return to its original path and continues following the new path.

It is well known that the principle of testing for (non)stationarity of time series is to determine whether a unit root (or a stochastic trend) is present in the analysed process. This is conducted by performing a unit root test, which is based on statistical hypothesis testing that tests a null hypothesis against an alternative one. The null hypothesis ($H_{0}$: ϕ=1) is that the time series has a unit root and, thus, is nonstationary; the alternative hypothesis ($H_{1}$: $\left|\phi \right|<1$) is that it does not have a unit root and so is stationary. The hypothesis test requires the calculations of unit root test statistics and critical values. The statistical hypothesis testing procedure can be described as follows:
If the absolute value of the calculated test statistic is greater than the critical value (in absolute value), then $H_{0}$ can be rejected (i.e., the time series is stationary).If the absolute value of the calculated test statistic is smaller than the critical value (in absolute value), then $H_{0}$ cannot be rejected (i.e., the time series is nonstationary).

In the field of econometrics, the Augmented Dickey–Fuller (ADF) test [21,22], Phillips–Perron (PP) test [23], Kwiatkowski–Phillips–Schmidt–Shin (KPSS) test [24], and Leybourne–McCabe (LM) stationarity test [25,26] are the most widely used unit root tests. In the following, these tests are shortly described.

### 2.4. Common Unit Root (or Stationarity) Tests

(a)Augmented Dickey–Fuller (ADF) test for a unit root

The ADF test uses the following model to assesses the null hypothesis of a unit root:(7)$$y_{t}=c+\delta t+\phi y_{t-1}+\beta_{1}\Delta y_{t-1}+\cdots +\beta_{p}\Delta y_{t-p}+\varepsilon_{t}$$
where c and δ are constants (c is the drift coefficient, and δ is the deterministic trend coefficient), Δ is the differencing operator (i.e., $\Delta y_{t}=y_{t}-y_{t-1}$), p is the number of lagged difference terms, and $\varepsilon_{t}$ is a mean zero innovation process.

The null hypothesis of a unit root is $H_{0}:\phi =1$; whereas the alternative hypothesis is $H_{1}:\phi <1$. If the ADF test rejects the null hypothesis, this means that the time series has no unit root and is stationary. An ordinary least squares (OLS) regression is employed in the ADF test algorithm to estimate the coefficients of the alternative model [22]. Some appropriate variants of the model include the following: (1) the model with c=0 and δ=0 has no drift and time trend, (2) the model with δ=0 has no trend component, (3) the model with c=0 has no drift component, and (4) the model with c≠0 and δ≠0 has both drift and time trend.

Three different cases can be considered for ADF tests, as described in the following. The name of the case is represented by the type of alternative model.

Case I—autoregressive (AR) model variant: specifying an ADF test of the null model, given by Equation (7) with c=0, δ=0, and ϕ=1, against the alternative model, given by Equation (7) with c=0 and δ=0 and with coefficient ϕ<1.

Case II—AR model with drift variant: specifying an ADF test of the null model, given by Equation (7) with c=0, δ=0, and ϕ=1, against the alternative model, given by Equation (7) with c≠0, δ=0, and ϕ<1.

Case III—Trend stationary model variant (or AR model with drift and time trend variant): specifying an ADF test of the null model, given by Equation (7) with c≠0, δ=0, and ϕ=1, against the alternative model, given by Equation (7) with c≠0, δ≠0, and ϕ<1.

(b)Phillips–Perron (PP) test for a unit root

The Phillips–Perron unit root test uses the model given by the following [23]:(8)$$y_{t}=c+\delta t+\phi y_{t-1}+\varepsilon_{t}$$
where the drift coefficient c and the deterministic trend coefficient δ are constants, and $\varepsilon_{t}$ is an innovation process.

The PP test tests the null hypothesis of a unit root (i.e., $H_{0}:\phi =1$) against the alternative ($H_{1}:\phi <1$). The test assesses the null hypothesis by using the model that is most appropriate for data with different characteristics of drift and time trend. A least squares regression is used to estimate coefficients in the null model. In addition, it uses modified Dickey–Fuller statistics [21,22] to confront serial correlations in the innovation process $\varepsilon_{t}$. Three different cases can be considered, as described in the following. The name of the case reflects the type of alternative model.

Case I—autoregressive (AR) model: The PP test tests the null model $y_{t}=y_{t-1}+\varepsilon_{t}$ against the alternative model $y_{t}=\phi y_{t-1}+\varepsilon_{t}$ with the coefficient ϕ<1.

Case II—AR model with drift: The PP test tests the null model $y_{t}=y_{t-1}+\varepsilon_{t}$ against the alternative model of the form $y_{t}=c+\phi y_{t-1}+\varepsilon_{t}$ with coefficient ϕ<1 and the drift coefficient c≠0.

Case III—AR model with drift and time trend (or trend stationary model): The PP test tests the null model of the form $y_{t}=c+y_{t-1}+\varepsilon_{t}$ against the alternative model $y_{t}=c+\delta t+\phi y_{t-1}+\varepsilon_{t}$ with coefficient ϕ<1, the drift coefficient c≠0, and deterministic trend coefficient δ≠0.

(c)Kwiatkowski–Phillips–Schmidt–Shin (KPSS) test for stationarity

The KPSS test assesses the null hypothesis of a trend stationary time series against the alternative of a nonstationary unit root process. The test is based on the following model [24]:(9)$$\begin{array}{l} y_{t}=c_{t}+\delta t+\varepsilon_{1t}\\ c_{t}=c_{t-1}+\varepsilon_{2t} \end{array}$$
where $c_{t}$ is a random walk time series, δ is the deterministic trend coefficient, $\varepsilon_{1t}$ is a stationary process, and $\varepsilon_{2t}$ is an independent and identically distributed process with mean zero and variance $\sigma^{2}$, i.e., $\varepsilon_{2t}\sim$i.i.d.($0,\sigma^{2}$).

The null hypothesis of the KPSS test is $\sigma^{2}=0$, meaning that the random walk term $c_{t}$ is constant and takes the role as the model intercept. The alternative hypothesis is that $\sigma^{2}>0$, implying that a unit root exists in the random walk series $c_{t}$. The KPSS test algorithm uses an OLS regression to fit between the data and the null model.

The KPSS test statistic is calculated as follows [24]:(10)$$\sum_{t=1}^{N}S_{t}^{2}{\left(s^{2}N^{2}\right)}^{-1}$$
where N is the sample size, $s^{2}$ is the estimate of the long-run variance, and $S_{t}=e_{1}+e_{2}+\cdots +e_{t}$ with $e_{t}$ is an innovation process.

(d)Leybourne–McCabe (LM) test for stationarity

The Leybourne–McCabe test is based on the following model [25]:(11)$$\begin{array}{l} y_{t}=c_{t}+\delta t+\beta_{1}y_{t-1}+\cdots +\beta_{p}y_{t-p}+\varepsilon_{1t}\\ c_{t}=c_{t-1}+\varepsilon_{2t} \end{array}$$
where $c_{t}$ is a random walk time series, δ is the deterministic trend coefficient, p is the number of lagged values of $y_{t}$ to include in the model, and $\varepsilon_{1t}$ and $\varepsilon_{2t}$ are independent and identically distributed random processes with mean zero, i.e., $\varepsilon_{1t}\sim$i.i.d.($0,\sigma_{1}^{2}$) and $\varepsilon_{2t}\sim$i.i.d.($0,\sigma_{2}^{2}$), where $\varepsilon_{1t}$ and $\varepsilon_{2t}$ are independent of each other.

The model in Equation (11) is second-order equivalent in moments relative to an autoregressive integrated moving average model, denoted as ARIMA(*p*,1,1), which has the following form [25]:(12)$$(1-L)y_{t}=\delta +\beta_{1}(1-L)y_{t-1}+\cdots +\beta_{p}(1-L)y_{t-p}+(1-aL)v_{t}$$
where *L* is the lag operator (i.e., $Ly_{t}=y_{t-1}$) and $v_{t}\sim$i.i.d.($0,\sigma^{2}$).

The null hypothesis of the LM test is that $\sigma_{1}^{2}=0$ and $\sigma_{2}^{2}=0$ (concerning the model in Equation (11)), which is equivalent to a=1 in the ARIMA model given by Equation (12). The alternative hypothesis is that $\sigma_{1}^{2}>0$ and $\sigma_{2}^{2}>0$, which is equivalent to a<1. Under the null hypothesis, the model in Equation (11) is an AR(*p*) model with an intercept $c_{t}$ and a deterministic trend *δt*; while the model in Equation (12) is an over-differenced ARIMA(*p*,1,1) representation of the same process. The LM stationarity test algorithm calculates test statistics in two steps [25]: first, it determines the maximum likelihood estimates of the coefficients in the ARIMA model and then regresses the filtered data $z_{t}=y_{t}-\beta_{1}y_{t-1}-\cdots -\beta_{p}y_{t-p}$ on an intercept $c_{t}$.

In this paper, the theoretical background of time series stationarity, testing for unit root, and the four tests used are briefly introduced. For more theoretical details, potential readers are referred to publications from the econometrics literature [18,19,20].

## 3. Lamb Wave Based Structural Damage Detection Using Unit Root (or Stationarity) Tests

Structural damage detection has been broadly studied for a wide range of aerospace, civil, and mechanical structures by applying various model-based and data-driven approaches. In essence, model-based methods aim at identifying damage by fitting a numerical model to real data, whereas data-driven techniques are based on processing data obtained from the monitored structure without relying on a priori models. As data measurements generally form multivariate sequences ordered by time, it follows that, in essence, a data-based SHM process is just a matter of time series analysis [29]. Furthermore, experimental or operational data representing a normal operating condition of the monitored structure or process are often assumed to be (relatively) stationary [29,30,31]. Another assumption is that damage or fault can potentially change or diminish the stationarity of data [8,28]. Different severities of damage or fault may also result in different stationary characteristics. Therefore, the analysis of stationarity of experimental and/or operational data has been employed in this research for damage detection and fault diagnosis.

In the context of this study, since the methodology proposed for damage detection is based on stationary statistical characteristics of Lamb wave data (presented in the form of time series), an important question that is raised is how do we measure the stationarity of a Lamb wave signal? The answer for this question is that, in principle, the degree of stationarity (or nonstationarity) of a signal or a time series can be quantified by using a unit root test [18]. This has been confirmed for the case of ADF test in the previous investigations [8,27,28,29,30,31], which can be stated as “the more negative the ADF t-statistics are, the more stationary the analysed data are”. Following the approach, we have proposed a general stationarity calculation procedure for Lamb wave based structural damage evaluation in this paper, as shown in Figure 6. In essence, the unit root (or stationarity) test is applied to Lamb wave data to calculate their t-statistics, which are then used as damage-sensitive indicators. The resulting t-statistics, exhibiting quantitatively the stationarity of Lamb wave responses, are used for detecting abnormal problems in the structure of interest. In this investigation, the four unit root testing methods, described in Section 2.4, are considered for the stationarity calculation procedure.

Due to the fact that the results of unit root tests on Lamb wave data are simply given in the form of decimal numbers, one can directly compare the t-statistics calculated for data acquired from different damage conditions of the structure. The difference or separation between t-statistic features can exactly reveal the state or condition of the structure. The average separation between t-statistics can also be calculated to provide a quantitative damage indicator. In this study, the stationarity calculation procedure for Lamb wave based structural damage detection was implemented by using the MATLAB Econometrics Toolbox™ [34].

## 4. Lamb Wave Experimental Data Contaminated by Varying Temperature

Lamb wave experimental data used in this study originate from a series of experiments described in [35]. The experiments used a pitch-catch configuration of Lamb wave propagation. The specimen was an aluminium plate of 2 mm thickness, had a rectangular shape (200 × 150 mm), instrumented two low-profile, and had surface-bonded piezoceramic Sonox P155 transducers (diameter 10 mm, thickness 1 mm). One transducer was used for Lamb wave generation and another one for response sensing. A five-cycle 75 kHz cosine burst signal was used as an excitation signal for the generation of Lamb waves. The excitation signal was enveloped by using a half-cosine function. The maximum peak-to-peak amplitude of excitation was chosen at 10 V. The experimental setup is illustrated in Figure 7. A TGA 1230 arbitrary waveform generator was used to generate excitation signals. A digital 4-channel LeCroy LT264 oscilloscope was used to acquire Lamb wave responses. In order to obtain Lamb wave data for various temperature levels, the instrumented aluminium plate was placed in a 100 L LTE Scientific oven during the experimental process. The temperature on the surface of the plate was measured by using a thermal probe.

The experiments were firstly performed by using the intact plate. The specimen was heated from 35 °C to 70 °C and then cooled from 70 °C to 35 °C, with a change step of 5 °C. This entire process was repeated one more time in order to address the repeatability issue and to check for possible hysteresis between heating and/or cooling steps. Subsequently, a hole of 3 mm diameter was drilled in the middle of the same specimen, and the entire experimental investigation was repeated to collect Lamb wave data for the damaged case. It is noted that the acquisition of Lamb wave responses was repeated 50 times at each temperature level. In other words, we obtained 50 Lamb wave measurements for each damage-temperature case. In total, 1450 measurements were recorded at 29 temperature levels for each intact or damaged case using the sampling rate of 10 MHz; 5000 data samples were acquired for each measurement.

This study has used Lamb wave data acquired at four different temperatures (35 °C, 45 °C, 60 °C, and 70 °C) for both intact and damaged cases. This forms an experimental case study of Lamb wave based SHM under temperature variations. More specifically, four temperature-dependent data sets for each damage condition have been analysed. Each data set at a certain temperature level consists of 20 Lamb wave responses, which are formed by randomly selecting five measurements at each heating or cooling phase. For example, the first data set for the intact plate at 35 °C consists of five measurements at the first heating step, five measurements at the first cooling step, five measurements at the second heating step, and five measurements at the second cooling step. Varying temperatures have caused strong effects on the investigated Lamb wave data, as reported previously in [35] and recently in [8]. Figure 8 and Figure 9 provide examples of Lamb wave responses at different temperature levels for intact and damaged cases, respectively. The first wave packet in all plots is a symmetrical Lamb wave mode. The successive wave packets are reflections from the boundaries and damage (in the case of Figure 9). It can be observed from Figure 8 and Figure 9 that the amplitudes and shapes of Lamb wave responses slightly changed due to both damage and temperature. These responses consist of different wave packages (dispersed incident waves, reflected components, and scattered components of two wave modes, i.e., A0 and S0 Lamb wave modes). These responses are nonstationary by definition; statistical properties change over time. All responses change due to two effects, i.e., structural damage and due to temperature. Figure 8 and Figure 9 demonstrate that these effects are not clearly visible and are difficult to separate. Both effects introduce a trend to these responses. However, it is hard to recognize any clear trends in the data. It was discussed in [36] that varying temperatures could affect surface bonding (adhesion) between transducers and the structure, resulting in the variations in both the amplitude and the phase of the signal. Recently, the work in [37] has provided a comprehensive review of the effects of environmental and operational conditions (EOCs) on Lamb wave propagation in various structures. A number of efficient strategies for the compensation and/or removal of EOCs effects for Lamb wave based SHM systems have been highlighted in the review.

## 5. Results and Discussion

This section presents an application of the proposed structural damage detection method using unit root or stationarity tests (presented in Section 3) for Lamb wave experimental data contaminated by varying temperatures (described in Section 4). Four different unit root tests, i.e., ADF, KPSS, PP, and LM test, have been used to assess the stationarity of Lamb wave data before and after the damage introduced to the specimen. The capability of the analysed tests for damage detection is compared and discussed.

The selected results obtained for four different temperature-dependent data sets at 35 °C, 45 °C, 60 °C, and 70 °C are shown in Figure 10, Figure 11, Figure 12 and Figure 13, respectively. At first glance, one can observe that the results are relatively identical for all investigated temperature levels. Regarding both ADF test and KPSS test, almost all t-statistics of the undamaged data are very well separated from t-statistics of the damaged data. Furthermore, the t-statistic features representing the undamaged case are much more negative than the t-statistic features representing the damaged case. As discussed in Section 3, this behaviour was reported for the case of ADF test in the previous work [8,27,28,29,30,31]. In this study, it is interesting that the KPSS test demonstrates the same behaviour. Thus, for both ADF and KPSS tests, the more negative the t-statistic features, the more stationary the analysed Lamb wave data are. This confirms the fact that Lamb wave data measured from the intact plate are more stationary than the data measured from the damaged plate. It is obvious that the decline of the degree of stationarity with respect to the damaged data is due to the damage introduced to the plate. In contrast to the results of ADF and KPSS tests, there are many overlaps between t-statistics of the undamaged and damaged data with respect to both PP test and LM test. Therefore, the first conclusion that can be drawn from the plots in Figure 10, Figure 11, Figure 12 and Figure 13 is that both ADF test and KPSS test can detect the 3 mm hole damaged case, whereas both PP test and LM test are not significant for identifying the damage.

Next, it is important to compare the damage detection results between ADF test and KPSS test. A quantitative comparison has been performed in order to provide a convincing assessment. Although both tests produce divergence between t-statistics of the intact and damaged data sets for all temperature levels investigated, the separations seem to be much smaller in case of ADF tests. In order to demonstrate quantitative comparison between ADF test and KPSS test more clearly, the separations of t-statistics between the intact and damaged data are calculated in the form of the absolute value (or modulus) at each Lamb wave response. Then, the average of 20 absolute values of separations for each temperature case is calculated. The calculations have been performed for all t-statistic features shown in Figure 10, Figure 11, Figure 12 and Figure 13 (only for the ADF and KPSS t-statistics). The results are shown in Table 1. It can be observed that the ADF test creates (almost) the same level of separation at all investigated temperatures, whereas KPSS test provides much larger separation, however, with values varying with temperature. In particular, the degree of separation reduces significantly at 70 °C. The results imply that the ADF test is more stable with respect to temperature changes than the KPSS test. However, the KPSS test can detect damage better than the ADF test.

It is important to provide a further discussion about the influence of temperature on the stability of KPSS and ADF test statistics. First, regarding the t-statistics obtained for the undamaged case, it can be observed from Figure 10, Figure 11, Figure 12 and Figure 13 that all features have the same pattern, that is, irregular and unpredictable fluctuations. This behaviour of the t-statistics can be caused by the influence of temperature (i.e., the dominant factor) as well as the other less dominant factors such as vibration, humidity, noise, varying experimental conditions, etc. Second, with respect to the damaged case, interestingly, the KPSS t-statistics calculated for the temperature-dependent data set at 35 °C (see Figure 10) exhibited almost the same level of stationarity, which is indicated by the relatively stable t-statistics around the value of −25 at all 20 Lamb wave responses. The same results can be observed in Figure 11 and Figure 12 for the temperature-dependent data sets at 45 °C and 60 °C, respectively. However, the KPSS t-statistics obtained for Lamb wave data of the damaged case at 70 °C show fluctuating behaviour and are less stable, as shown in Figure 13. It is noted that KPSS t-statistics start fluctuating for the damaged data at 65 °C with respect to the influence of temperature on the ADF t-statistics obtained for the damaged case. For temperatures at 35 °C, 45 °C, and 60 °C, ADF t-statistics were relatively stable within the value of −9 and −11 at all 20 Lamb wave responses, as shown in Figure 10, Figure 11 and Figure 12. However, the fluctuating range of ADF t-statistics becomes much larger at 70 °C, i.e., between −5 and −12. These results imply that both KPSS test and ADF test can consistently detect the damage in the conditions of temperature varying below 60 °C. However, their t-statistics fluctuate more (or less homogeneous) for temperatures higher than 65 °C.

All Lamb wave responses consist of multiple wave packages and are nonstationary. This nonstationary behaviour is exhibited in Figure 8 and Figure 9. However, these responses are corrupted by trends—that increase the level of stationarity—due to damage and environmental effects. The former relates to well known physical phenomena—i.e., attenuation of the incident wave, scattering of the incident wave, and interaction of the incoming incident wave and wave reflected from damage—as explained for the example in [38]. The latter also relates to physics; wave velocity (and, therefore, Lamé constants) changes due to temperature, and wave amplitude is also affected due to the piezoelectric effect of transducers. However, the former (damage-related) is a local effect (does not matter whether caused by a crack or a hole) whereas the latter (temperature-related) is a common trend that is accumulated. It appears from the tests of stationarity that the damage-related trend contributes more to nonstationarity and, therefore, can be used for damage detection.

It should be remarked that the results presented in this paper are representative for the all Lamb wave data sets investigated. The results obtained for other temperatures were very similar with the results presented here.

## 6. Conclusions

Lamb waves are associated with a complex wave propagation mechanism that makes the directly analysis and interpretation of Lamb wave propagation modes for information about the health conditions of a structure a nontrivial task. Therefore, instead of directly analysing and interpreting Lamb wave responses for damage detection, this study has proposed a new approach, which is based on the analysis of stationary statistical characteristics of Lamb waves. The method has been experimentally validated by using Lamb wave data, contaminated by temperature variations, from intact and damaged aluminium plates. Four different unit root tests, such as ADF, KPSS, PP, and LM tests, have been investigated to make statistical inference about the stationarity of Lamb wave data before and after a hole damage is introduced to the aluminium plate. The separation between t-statistic features, obtained from the unit root tests on Lamb wave data, is used for damage detection. Based on the results obtained, some conclusions are given as follows:Both ADF and KPSS tests can detect the damage, while both PP and LM tests are not significant for identifying the damage.The ADF test is more stable with the temperature changes than KPSS test. However, the KPSS test can detect damage better than the ADF test.Both KPSS and ADF tests can consistently detect damage in the conditions of temperatures varying below 60 °C. However, their t-statistics fluctuate more (or less homogeneous) for temperatures higher than 65 °C.Based on these results, both ADF and KPSS tests are suggested to be used together for Lamb wave based structural damage detection in order to enhance the accuracy and reliability of the proposed stationarity-based approach.

This is a feasibility study, and the results presented in this paper are preliminary; therefore, further work that uses different types of specimens, more complex structures, and real damages (e.g., fatigue cracks in metals and delamination in composites) is required in order to confirm these findings. Moreover, the qualitative and quantitative comparisons of the proposed method with other existing techniques will be investigated in the future. Moreover, research on applying unit root tests for Lamb wave based damage classification to distinguish different damage severities has been planned.

Since the proposed method is based on the concept of stationarity of time series, it can be applied for not only Lamb wave based SHM but also condition monitoring and fault diagnosis of industrial systems if the measured data can be presented in the form of time series. For example, vibration data of rotating machines, SCADA data of wind turbines, and vibration responses (or natural frequencies) of bridges would be suitable for analysis by this method.

## Figures and Tables

**Figure 1 materials-14-06823-f001:**
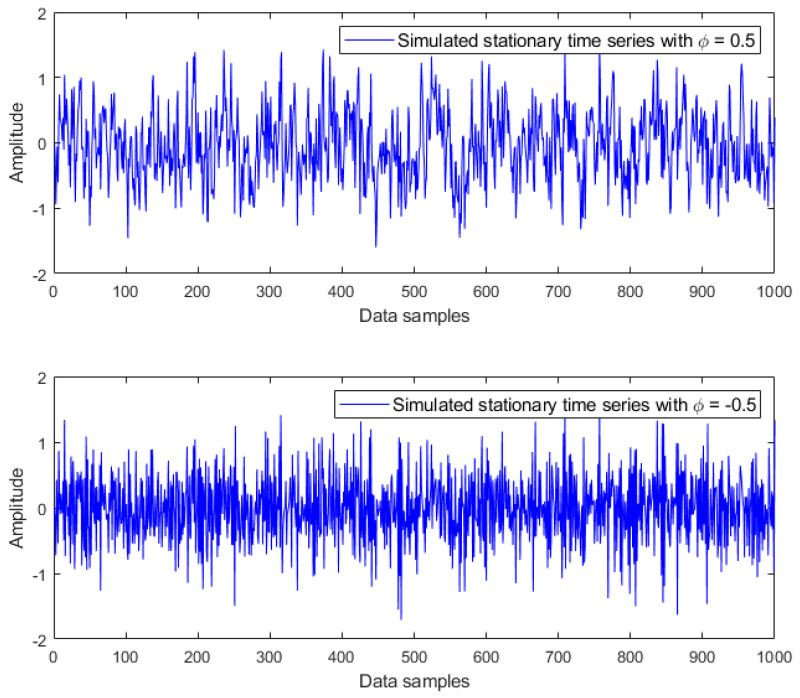
Simulated stationary time series with coefficients ϕ=0.5 and ϕ=−0.5.

**Figure 2 materials-14-06823-f002:**
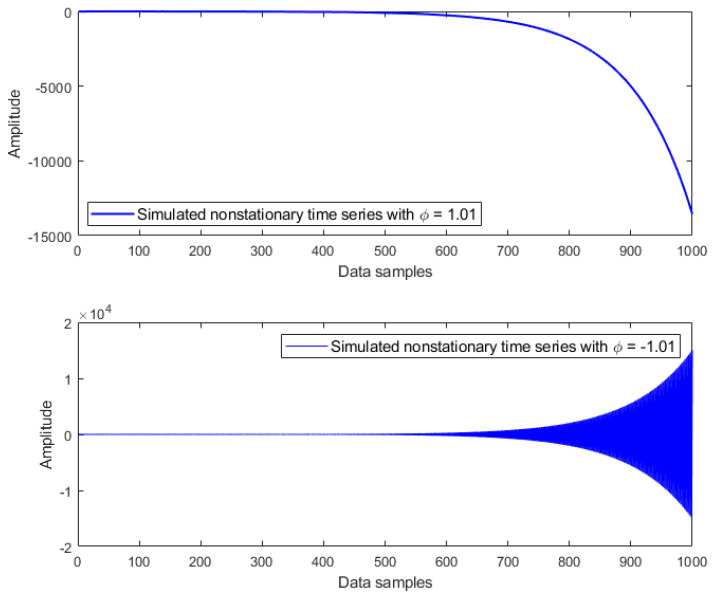
Simulated nonstationary time series with coefficients ϕ=1.01 and ϕ=−1.01.

**Figure 3 materials-14-06823-f003:**
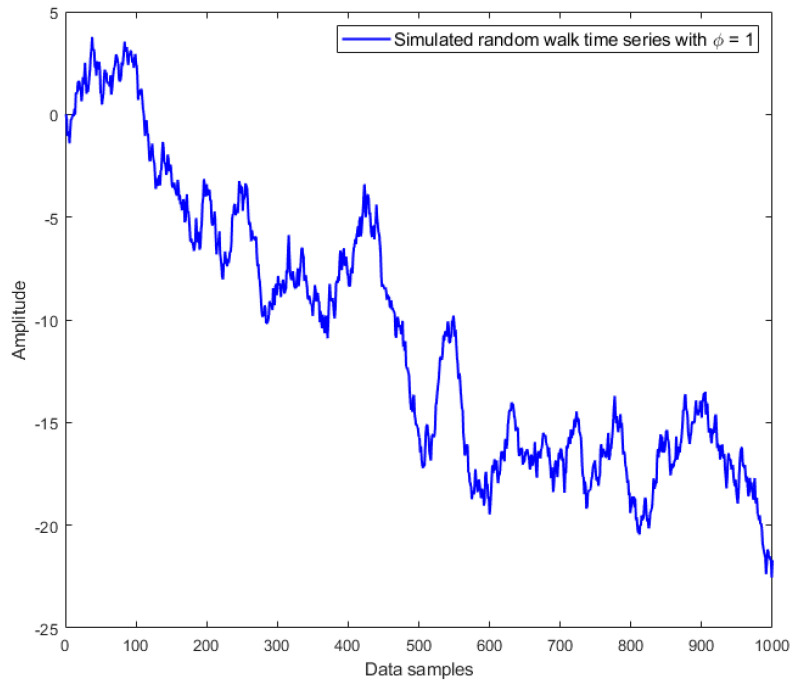
Simulated random walk time series with coefficient ϕ=1.

**Figure 4 materials-14-06823-f004:**
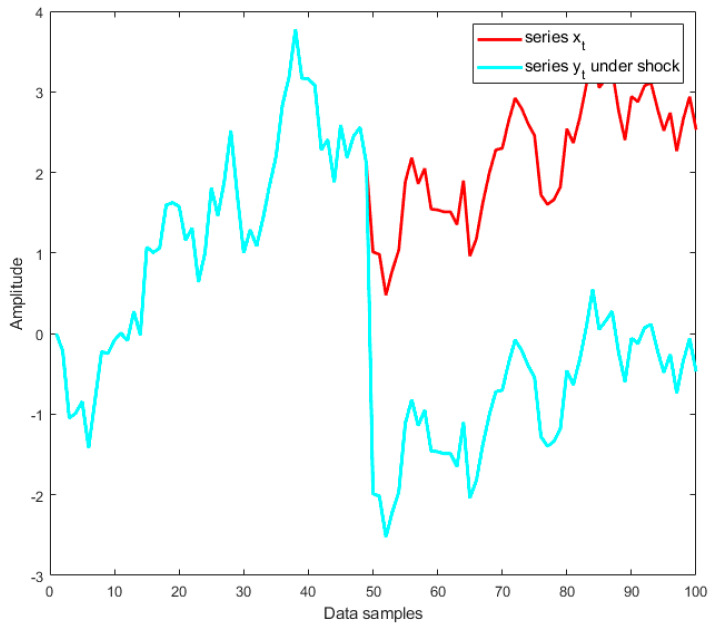
Shock to a stochastic trend (or a unit root) process.

**Figure 5 materials-14-06823-f005:**
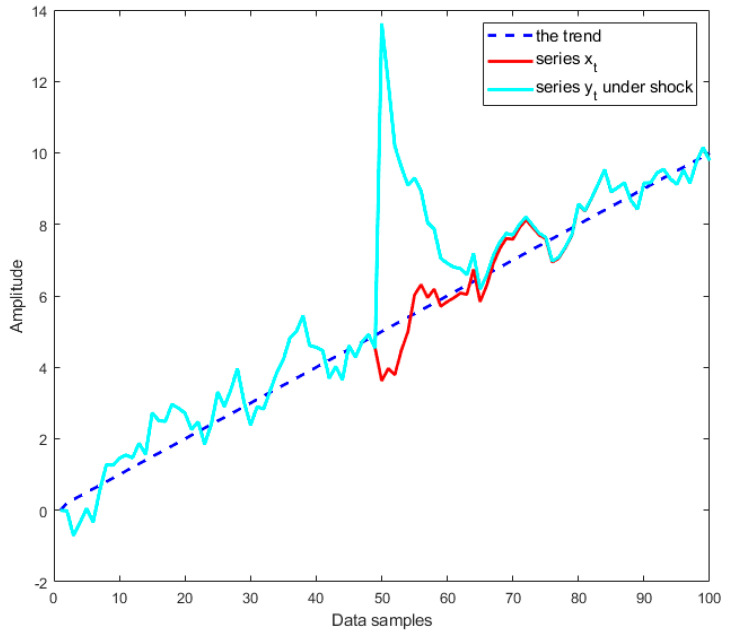
Shock to a deterministic linear trend (or trend-stationary) process.

**Figure 6 materials-14-06823-f006:**
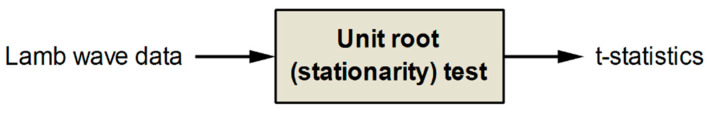
Stationarity calculation procedure for Lamb wave based structural damage detection.

**Figure 7 materials-14-06823-f007:**
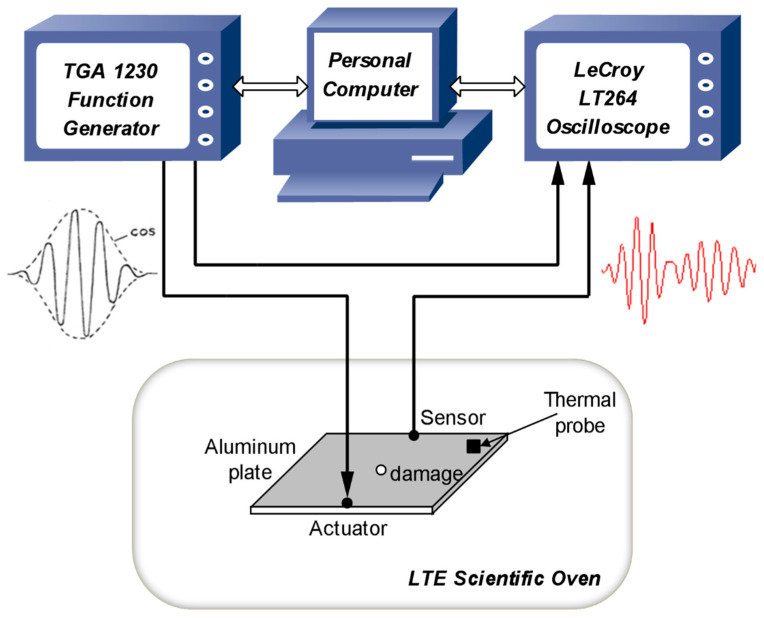
Schematic of the experimental setup used for Lamb wave data.

**Figure 8 materials-14-06823-f008:**
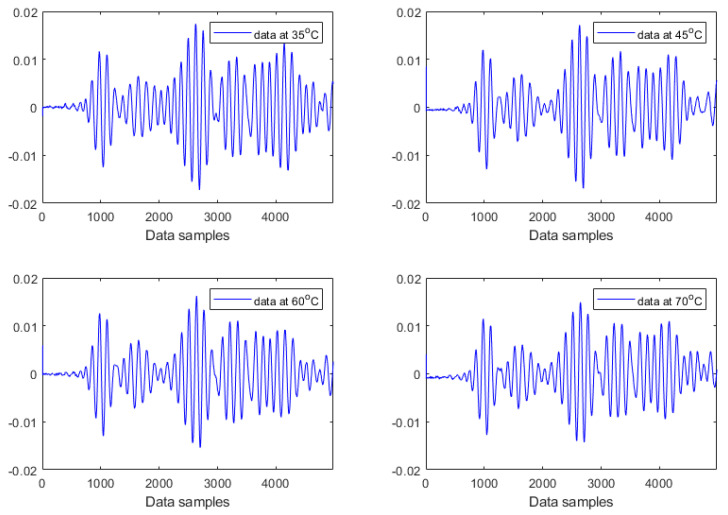
Examples of Lamb wave responses acquired from the intact plate for different temperatures.

**Figure 9 materials-14-06823-f009:**
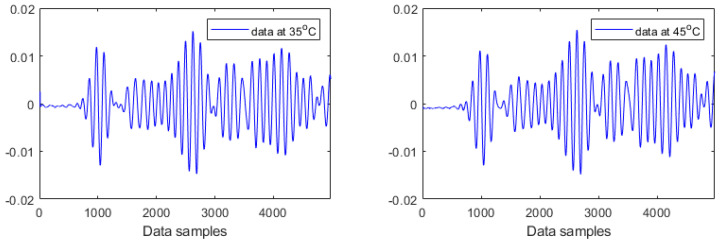
Examples of Lamb wave responses acquired from the damaged plate for different temperatures.

**Figure 10 materials-14-06823-f010:**
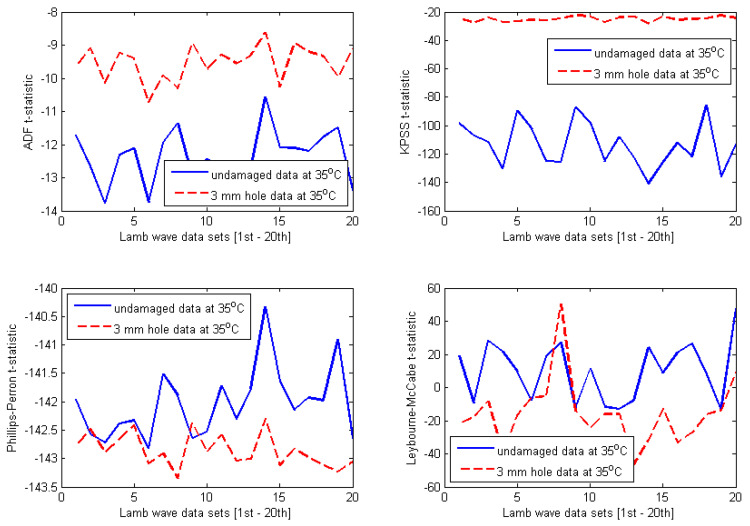
Results obtained for Lamb wave data at 35 °C.

**Figure 11 materials-14-06823-f011:**
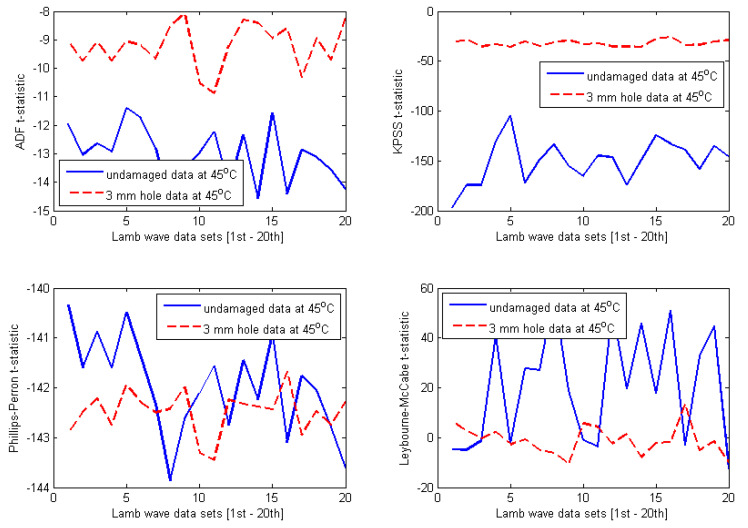
Results obtained for Lamb wave data at 45 °C.

**Figure 12 materials-14-06823-f012:**
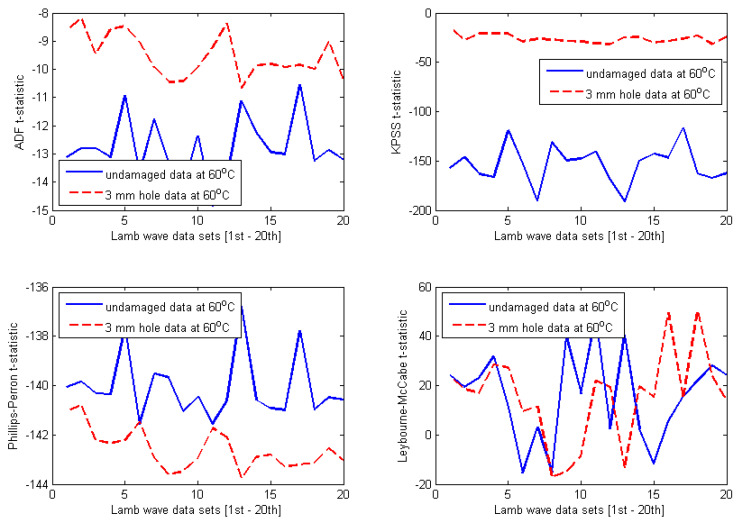
Results obtained for Lamb wave data at 60 °C.

**Figure 13 materials-14-06823-f013:**
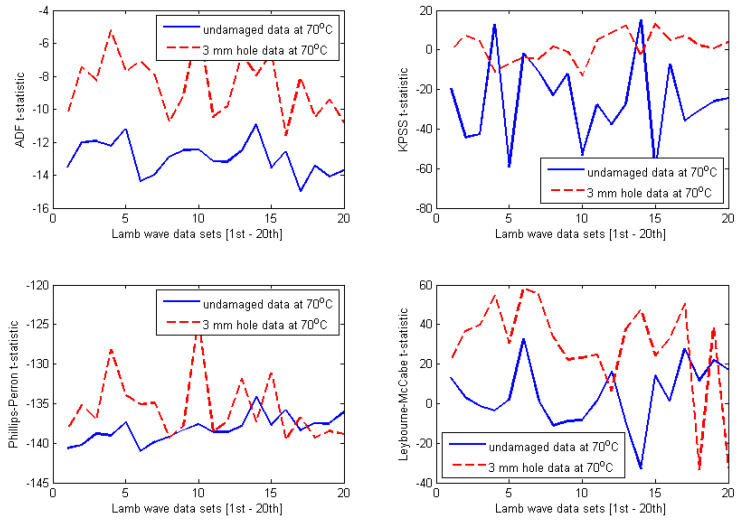
Results obtained for Lamb wave data at 70 °C.

**Table 1 materials-14-06823-t001:** Average separation between t-statistics of the undamaged and damaged data at different temperature cases.

Type of Tests	Data at 35 °C	Data at 45 °C	Data at 60 °C	Data at 70 °C
ADF test	2.81	3.83	3.28	4.42
KPSS test	88.43	118.21	127.62	31.63

## Data Availability

No new data were created in this study. Data sharing is not applicable to this article.

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
