# Peer review of "Lamb Wave Based Structural Damage Detection Using Stationarity Tests"

_materials, 2021, doi:10.3390/ma14226823_

Round 1

Reviewer 1 Report

see file attached

Reviewer 2 Report

Major revisions

Reviewer 3 Report

The author presented a state-of-the-art approach to detecting damages in thin plates based on the statistical analysis of stationarity (or nonstationarity) of Lamb waves. The work was well presented and the results were well analyzed and discussed. 

Reviewer 4 Report

  1. There are too many grammatical problems in the written English that should be improved before the acception;
  2. The author should clealy state the application of the proposed method in the different types of damage;
  3. In the experimental verification, the surface morphology and the specific shape of the designed damage should be presented;
  4. The figures can be improved to better illustrate the results;
  5. The errors and the efficiency of the proposed method should be analyzed by comparison with other methods;
  6. The conclusions are suggested to be itemized for better compresion.

Round 2

Reviewer 4 Report

Accpetance for publication is suggested with the expection of their future work.